# Kaizen: Decomposing cellular images with VQ-VAE

**Daniel Majoral**[ID]*, **Marharyta Domnich**[ID]

Institute of Computer Science, University of Tartu, Tartu, Estonia

* danielmajoral@gmail.com

## Abstract

A fundamental problem in cell and tissue biology is finding cells in microscopy images. Traditionally, this detection has been performed by segmenting the pixel intensities. However, these methods struggle to delineate cells in more densely packed micrographs, where local decisions about boundaries are not trivial. Here, we develop a new methodology to decompose microscopy images into individual cells by making object-level decisions. We formulate the segmentation problem as training a flexible factorized representation of the image. To this end, we introduce Kaizen, an approach inspired by predictive coding in the brain that maintains an internal representation of an image while generating object hypotheses over the external image, and keeping the ones that improve the consistency of internal and external representations. We achieve this by training a Vector Quantised-Variational AutoEncoder (VQ-VAE). During inference, the VQ-VAE is iteratively applied on locations where the internal representation differs from the external image, making new guesses, and keeping only the ones that improve the overall image prediction until the internal representation matches the input. We demonstrate Kaizen's merits on two fluorescence microscopy datasets, improving the separation of nuclei and neuronal cells in cell culture images.

## 1 Introduction

Recent advances in imaging techniques have improved the quality and quantity of medical image data. At the same time, deep learning algorithms have opened new possibilities for automatic medical image analysis [1]. However, health research and medical diagnosis require near perfection since marginal errors can lead to colossal harm or even death. To aggravate things, biological structures can be very challenging with small overlapping individual elements of complex shapes. Thus, current approaches [2,3] for image analysis tend to miss objects. In contrast, humans seamlessly recognize all the different objects that compose a given medical image.

Thus, a potential avenue for advancing medical image analyses lies in exploring aspects of human perception that diverge from contemporary computer algorithms. For instance, the intricacies of human perception extend beyond a mere bottom-up process, not relying solely on sensory input. Instead, humans leverage acquired knowledge and experiences to

**Data availability statement:** All relevant data will be held in a public repository after acceptance: https://figshare.com/projects/Kaizen/238289

**Funding:** This work was supported by Revvity (https://www.revvity.com/). DM and MD received research support by the Estonian Research Council Grants PRG1604, the European Union's Horizon 2020 Research and Innovation Programme under Grant Agreement No. 952060 (Trust AI), by the Estonian Centre of Excellence in Artificial Intelligence (EXAI), and by the Estonian Ministry of Education and Research.

**Competing interests:** This work was supported by Revvity (https://www.revvity.com/). This does not alter our adherence to PLOS ONE policies on sharing data and materials. There are no patents, products in development or marketed products associated with this research to declare.

generate plausible internal hypotheses [4]. Such internally generated hypotheses are continuously evaluated against external inputs, rejecting the hypotheses that diverge more from reality. In essence, humans possess generative models to dynamically construct and refine their perception of reality.

The use of generative models for computer vision allows to optimize the output during inference. For example, the reader might consider the case when the optimal solution for an image during inference is available. In this scenario, we can optimize the output to match the optimal solution. Typically during inference, we do not possess the optimal solution. However, we possess the ground truth used in self-supervised learning, the data structure of the image itself. Thus, it is possible to optimize the model solution to the degree that the image contains information about the optimal solution.

Generating images that already exist in the input might seem inefficient. However, generating internal images allows to contrast the internal prediction with the input, providing continuos error feedback. The error feedback might be used to update the model priors in an unsupervised manner, but also to improve object prediction during inference. For example, any object predictions inconsistent with the input image can be eliminated, improving results. Furthermore failing to predict an object in a region leads to a high local error, allowing to generate new object predictions in the specific location.

Following this reasoning, we present *Kaizen* a practical implementation for decomposing cellular images into individual objects. First, we train a VQ-VAE to encode individual cells. Then, the VQ-VAE makes predictions about the input during inference at image locations with maximal error. Like an evolutionary algorithm, only the predictions that decrease the difference between the input image and global prediction survive.

## 2 Related work

### 2.1 VQ-VAE

Variational Autoencoders [5] can compress an image dataset into a latent multivariate gaussian distribution. For further image compression, quantization methods have been applied successfully [6,7]. Similarly, Van den Oord et al. proposed the Vector Quantised-Variational AutoEncoder (VQ-VAE) [8] that also encodes input data to a vector of discrete latent variables. However, the VQ-VAE employs the discrete latent variables as indexes to a memory table to recover a set of embeddings. The embeddings are decoded to produce an image. Posterior work [9] proposed a hierarchical VQ-VAE with several latent codes to improve image quality.

### 2.2 Cell instance segmentation

Segmentation is considered the first critical step for biomedical microscopy image analysis. Although standard deep learning approaches have been applied to the task [10,11], some proposals explore alternative data representations that might be more suitable for microscopy. For example, StarDist [12,13] segments cell nuclei by describing them as star-convex polygons. StarDist first predicts for each pixel the distance to the cell nucleus boundary along a set of predefined equidistant angles, and afterward performs non-maximum suppression (NMS) to discard duplicated polygons. Multistar [14] extends Stardist by identifying overlapping objects, while SplineDist [15] modifies StarDist by representing objects as planar spline curves. Another example is Cellpose [16,17] learns during training to predict the gradients of a diffusion process starting at the cell centre. Later during inference, Cellpose backtracks the predicted gradient to see which pixels converge to the same cell centre.

Finally, in amodal blastomere instance segmentation, a VQ-VAE has been used after a typical detection pipeline to generate mask representations from the image features [18]. In contrast, our method employs the VQ-VAE to generate individual objects representations and later performs the segmentation.

## 2.3 Image decomposition

Early work [19] already proposed image decomposition as an alternative to object detection pipelines. Other approaches tackle unsupervised learning of object representations through image reconstruction [20–23]. While MONet [20] and IODINE [22] apply a network recurrently, Slot-attention [21] and Capsule networks [23] apply an algorithm iteratively. Although *Kaizen* is similar since it iteratively applies a VQ-VAE, it is trained in a supervised manner. Unlike those models wich incorporate an explicit attention mechanism, *Kaizen* relies on a VQ-VAE to generate only individual objects. More recently Composer [24] decomposes an image into several decoupled representations including object instances, and then trains a diffusion model conditioned to these representations, to improve control on image generation.

## 3 Methodology

An Illustration of *Kaizen* is shown in Fig 1. *Kaizen* uses a VQ-VAE model trained on microscopy images to predict one individual cell from an image with multiple cells. During inference the VQ-VAE iteratively predicts individual cells in the input microscopy image (Fig 1A). *Kaizen* maintains an internal predicted image formed by all the cells predicted so far (Fig 1B). The difference between the internal predicted image and the external image is the error image (Fig 1C). *Kaizen* accepts a new prediction only when it reduces the error, making the external image and the internal prediction more similar. Furthermore, the new predictions are made on regions with higher error, avoiding duplicate predictions. The process is repeated until the method is unable to predict new cells. *Kaizen* components are described in more detail below.

### 3.1 VQ-VAE

As a generative model for *Kaizen* a VQ-VAE [8] was chosen, favouring inference speed over quality of the samples. The VQ-VAE was trained with small patches containing a few cells

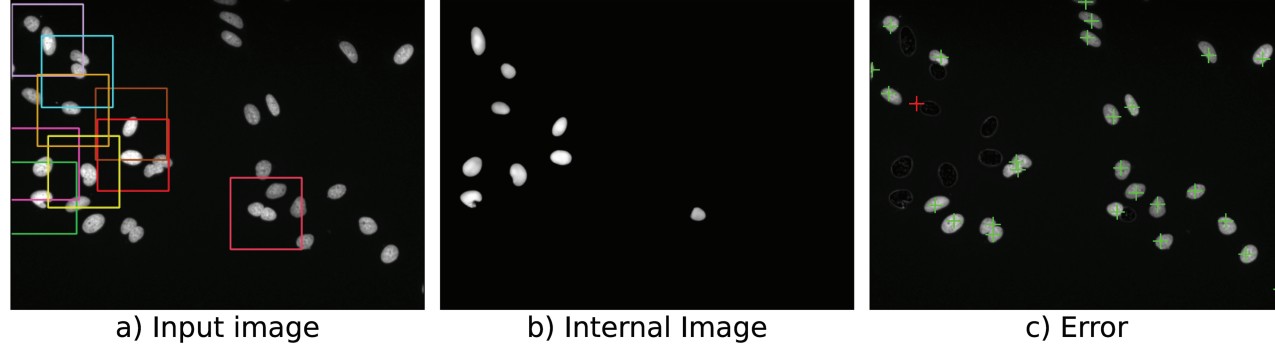

a) Input image     b) Internal Image     c) Error

**Fig 1. (a) Multicolored squares correspond to various regions serving as input to a VQ-VAE tasked with reconstructing the central cell within. The reconstructed individual cells are merged into an internal image, (b), with all the cells predicted so far. Subtracting this internal image from the original yields an error image, (c). New prediction points, indicated by crosses, are selected at local maxima in the error image. The prediction indicated by a red cross will be discarded since adding a cell there increases the overall error.**

from the hundreds in a microscopy image. The purpose of the training was for the VQ-VAE to produce a single cell image as output when given an image containing multiple cells. Thus, for a given input training patch the corresponding training output was generated by multiplying the input patch by the cell's mask touching the central pixel. As shown in Fig 2, after training the VQ-VAE encodes a single central cell from small patches containing multiple cells while disregarding other cells and noise present on the input. To avoid encoding non-existent cells, twenty percent of the training patches did not have any cell touching the central pixel; in these cases the VQ-VAE was trained to output an empty image.

## 3.2 Core algorithm

Using the VQ-VAE that encodes individual cells described above as predictor and taking inspiration from predictive coding in the brain, the core algorithm of *Kaizen* was implemented. Algorithm 1 calculates an error image as the difference between the input and a internal image prediction. At each iteration, several points are proposed where the error image reaches its maximum, and distant between them. Then at each proposed point in the image an object prediction is made. Finally, only predictions that diminish the loss when added to the internal prediction, are kept.

## 3.3 Predicting on the error

Notably, keeping an internal image allows estimation of an error image, which is derived by subtracting the internal prediction from the input image. Predicting on the error image filters out already predicted objects. Thus, predicting on the error image is easier, given its reduced complexity compared to the original image, minimizing potential confusion for the predictor.

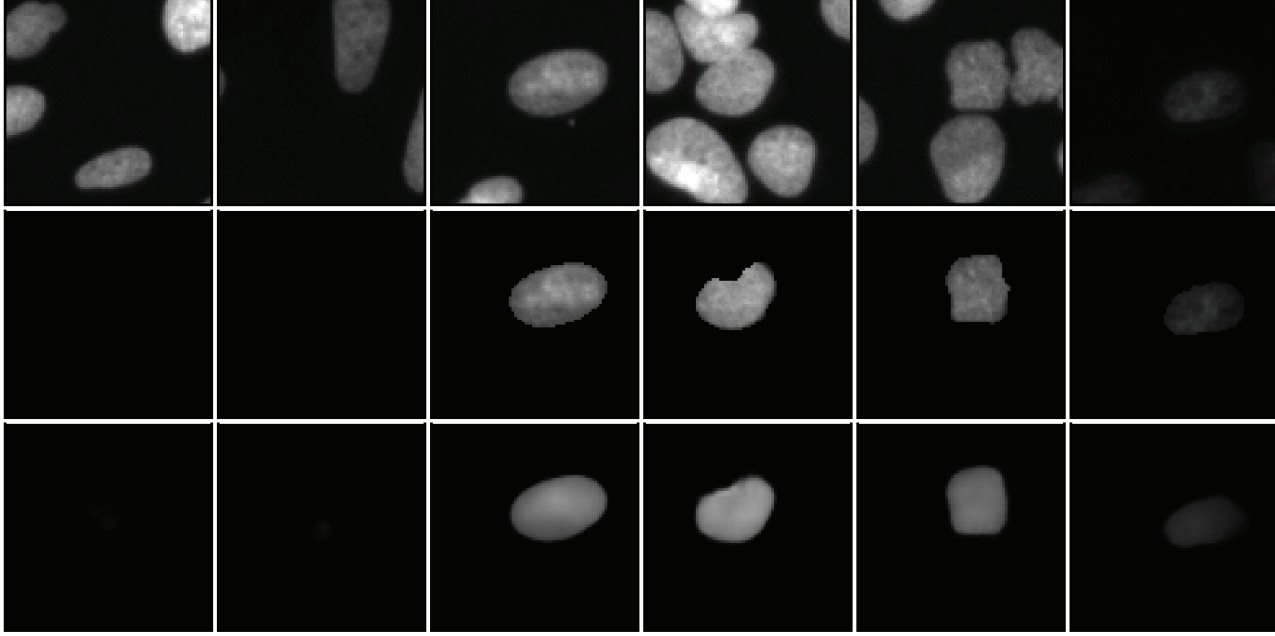

**Fig 2. Samples of the VQ-VAE encoding individual cells for the U2OS dataset.** The first row corresponds to training image patches employed as input to the VQ-VAE. In the second row, under each input, the corresponding ground truth is shown. Finally, the third row shows the corresponding VQ-VAE ouput after training.

**Algorithm 1. Core algorithm**

```
Initialize:
    internal ← zeroes
    loss_inter ← Loss(internal, image)
    error ← image
repeat
    {p₁, p₂, ..., p_N} = distant points of max(error)
    pred_p = predict at each point p in image
    loss_p = Loss(internal + pred_p, image)
    for any pred_p where loss_p < loss_inter do
        internal ← internal + pred_p
    end for
    loss_inter ← Loss(internal, image)
    error ← image - internal
until all loss_p ≥ loss_inter
```

However, erroneous predictions in the initial stages could propagate into subsequent ones, compounding any inaccuracies committed by the predictor.

To enhance object detection while minimizing error compounding, *Kaizen* first predicts on the input image with the core algorithm until no more object predictions are found. With this first set of predictions an error image is generated. Then, with the error image as input, we apply *Kaizen* again generating new predictions and a second error image. We repeat this process until no new objects are found in a given iteration, aiming to detect all objects even in the most densely populated images.

## 3.4 Datasets

U2OS dataset: Fluorescent microscopy images from U2OS cell lines. Image set BBBC039 version 1, available from the Broad Bioimage Benchmark Collection [25]. The ground truth consists of individual nucleus instances. Although the ground truth might contain some errors, we did not alter the ground truth. We randomly selected 100 images for the training set, 50 for the validation set, and 50 for the test set.

Neuroblastoma dataset: Fluorescent images of cultured neuroblastoma cells, available from the Cell Image Library [26]. The ground truth consists of manually annotated cell boundaries. Although the ground truth might contain some errors, we did not alter the ground truth. However, since the ground truth size was half the weight and height of the images, we shrink the images with cubic interpolation to match the ground truth size. We randomly selected 71 images for the training set, 12 for the validation set, and 17 for the test set.

## 3.5 Evaluation metric

A predicted object is considered a true positive (TP) if his intersection over union IoU with a ground truth object is above certain threshold $\tau$. For the same threshold predicted objects without ground truth are considered false positives (FP), and finally ground truth not predicted is considered false negative (FN). The average precision (AP) for one image is given by:

$$AP_\tau = \frac{TP_\tau}{TP_\tau + FN_\tau + FP_\tau}$$ (1)

The average precision reported is the average over all images in the test set.

### 3.6 Implementation details

The VQ-VAE encoder consists first of 5 strided convolutional layers with padding 1. The first three layers have stride 2 and window size $4 \times 4$. Fourth layer has stride 1 and window size $4 \times 4$. Fifth layer has stride 1 and window size $3 \times 3$. The first two layers have 64 hidden units while the rest of the layers have 128 hidden units. The convolutional layers are followed by two residual $3 \times 3$ blocks (implemented as ReLU, $3 \times 3$ conv, ReLU, $1 \times 1$ conv)

The VQ-VAE decoder first has 1 convolutional layer of stride 1, window size $4 \times 4$, padding 1 and 128 hidden units. Then it has two residual $3 \times 3$ blocks. Next, three transposed convolutions follow with stride 2, window size $4 \times 4$, and the first one with padding 1. The final layer is a transposed convolution with stride 1, window size $3 \times 3$ and 64 hidden units.

The VQ-VAE codebook contains 128 embeddings of dimension 2, and we use exponential moving averages [8] to update the dictionary with $\beta = 0.25$. We use the ADAM optimiser [27] with learning rate 1e–4, L1 loss, and train for 500,000 steps with batch-size 32. L1 loss was also employed for the *Kaizen* algorithm.

To predict at the border of the image, the input image was padded by half the input size of the VQ-VAE, eight pixels of mirror-padding followed by zero-padding (20 pixels in total for the U2OS dataset and 60 for the Neuroblastoma dataset).

To select several distant points in parallel, we convolve the error image with a $7 \times 7$ kernel of ones with stride one. The highest value in the resulting image is the first point. The region surrounding this point ($32 \times 32$ pixels) is set to zero, and the process is repeated to select subsequent points. This process aims to minimize the occurrence of predictions on background noise and cell boundaries.

For the U2OS dataset the number of simultaneous points of prediction was set to ten (N in Algorithm 1). To avoid a hypothetical infinite loop, the repeat loop in Algorithm 1 was set to a maximum of 30 iterations. For the neuroblastoma dataset the number of simultaneous points of prediction was set to one (N in Algorithm 1) and the repeat loop in Algorithm 1 was set to a maximum of 100 iterations.

As pre-processing the images of both datasets were normalized. No data augmentation was applied. As post-processing to avoid empty predictions in the U2OS dataset, we eliminate all the masks with less than 20 pixels. For the same reason, in the neuroblastoma dataset, all masks with less than 20 blue pixels or less than one green pixel are eliminated.

## 4 Results

*Kaizen* decomposes an image into object representations, including superpositions of objects or occlusions. In contrast, typical computer biomedical models are classifiers that predict box coordinates plus category classification (object detection) or perform pixel classification (segmentation). Thus, to compare the current methodology, our method was modified to perform instance segmentation. Specifically, the predicted objects are converted to binary masks by setting a minimum brightness threshold (ten percent of the input image average), such that all the pixels above it are set to one and below it to zero. Such an approach might understate *Kaizen* results but provides a reasonable comparison to the current methodology.

First, we evaluate *Kaizen* on a U2OS dataset of cell nuclei fluorescent microscopy images [15]. To train the VQ-VAE we generated image patches of $40 \times 40$ pixels, with eighty percent of them containing a cell touching the patch center. The VQ-VAE was then trained to code representations of individual cell nuclei as illustrated in Fig 2 and described in 3.1.*Kaizen* was then applied on 50 left-out images from the dataset. Qualitative results for a test image are illustrated in Fig 3.

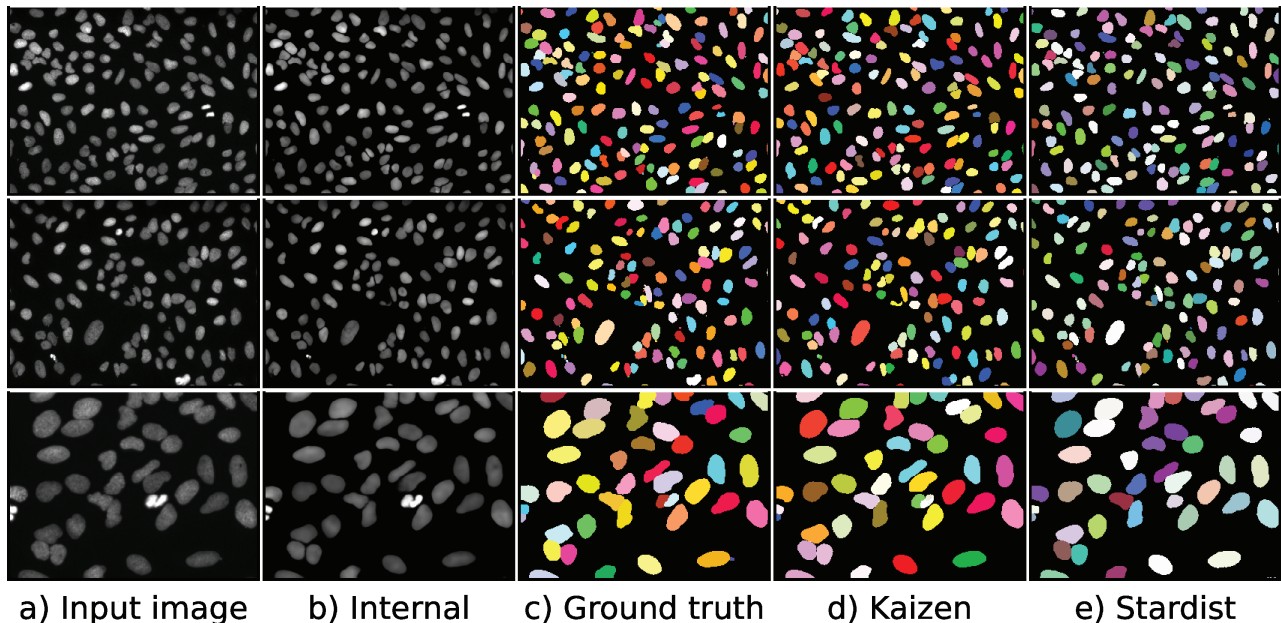

**Fig 3. Examples of *Kaizen* segmentation for the U2OS dataset.** The first two rows correspond to two dataset images, while the last row magnifies an image region. Panel a) depicts the original dataset image, and panel b) shows the corresponding internal image reconstruction created by merging the individual nuclei generated by the VQ-VAE. Panels d) and e) show Kaizen and Stardist corresponding image segmentation. Panel c) depicts the ground truth segmentation.

Regarding numerical results, we compare *Kaizen* to Stardist model [12], because it was specifically designed to predict cell nuclei in the same data type, and it is superior to more general approaches like U-Net [2] or Mask R-CNN [3]. The numerical evaluation in Table 1 shows that our method obtains superior average precision across all the thresholds.

On the whole U20S test set, Kaizen produces 100 false positives and 486 false negatives. Thus, the majority of Kaizen errors correspond to false negatives or ground-truth masks that have no valid match. Most of these non-predicted nuclei correspond to extremely minuscule nuclei, labeled inconsistently in the ground truth.

The impact on the algorithm of variations across entire images was also analyzed. As illustrated in Fig 4, with a fixed number of parallel predictions the processing time of Kaizen scales proportionally with the number of cells present in the image. For empty images, the computation is nearly instantaneous, as the algorithm primarily involves convolving a kernel of ones across the image and performing a limited number of predictions. Consequently, Kaizen remains highly efficient for large images with a low cell density. However, computational time

**Table 1. Results for the average precision(AP) for several intersection over union (IoU) thresholds for the U2OS nucli dataset and Neuroblastoma dataset.**

| Threshold | 0.5 | 0.6 | 0.7 | 0.8 | 0.9 |
|---|---|---|---|---|---|
| U2OS nuclei | | | | | |
| Stardist | 0.908 | 0.889 | 0.864 | 0.812 | 0.576 |
| Ours | **0.931** | **0.916** | **0.893** | **0.845** | **0.584** |
| Neuroblastoma dataset | | | | | |
| Cellpose | 0.657 | 0.612 | 0.574 | **0.516** | **0.364** |
| Ours | **0.850** | **0.802** | **0.685** | 0.476 | 0.190 |

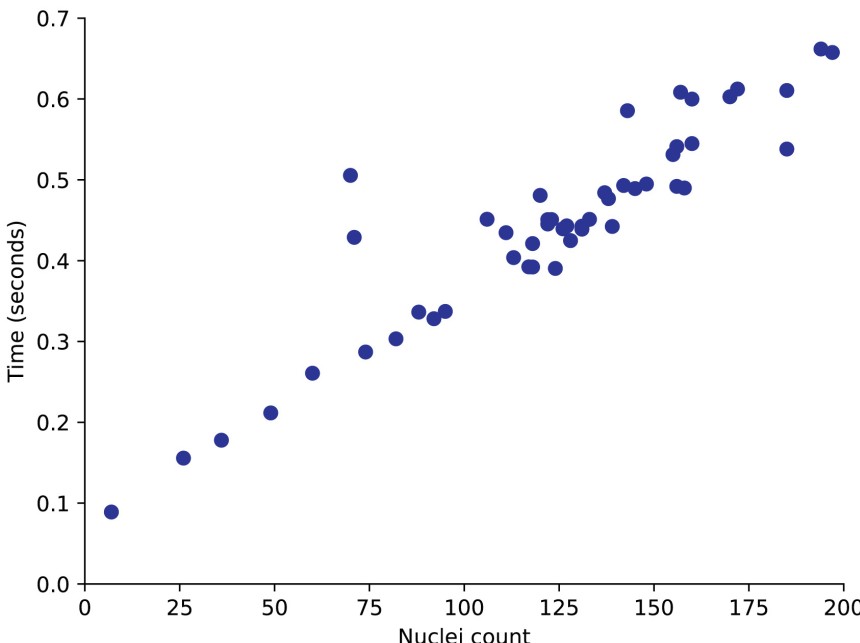

**Fig 4. *Kaizen* processing time versus the number of cells in the image for the U2OS test dataset at a fixed number of parallel predictions.**

increases in cases of high cell density, images with significant noise and artifacts, or instances where the model encounters cells that are challenging to predict.

Next, to assess *Kaizen* in more complex images, we evaluate it on a Neuroblastoma dataset of fluorescent microscopy images. In contrast to the U2OS dataset, the entire cell is predicted including the cytoplasm. To account for the increased individual prediction size, we increase the input size of the VQ-VAE to 120x120 pixels. *Kaizen* was applied on 17 left-out images from the dataset. Qualitative results for a test image are illustrated in Fig 5.

For the numerical results on the Neuroblastoma dataset, we compare *Kaizen* to Cellpose [16], since it was designed with this specific dataset. Numerical evaluation is presented in Table 1. Both models obtain worse results than expected; this might be the result of chance since the 17 left out images seem difficult compared to the typical image on the dataset. *Kaizen* obtains superior average precision for the three lowest thresholds. However, it falls behind for the 0.8 and 0.9 thresholds. Upon inspection, average precision decay for higher thresholds might be related to the conversion from object images to binary masks.

## 5 Conclusion

We have introduced a new approach to learning object representations in microscopy images: *Kaizen*. The approach is inspired by human perception, which contains an inherent predictive component that provides feedback to an internal model of the world.

In the implementation presented here, a VQ-VAE was trained to encode discrete representations of individual cells in microscopy images. Afterward, during inference, the VQ-VAE was applied iteratively on the input image to make new guesses, keeping only those that diminish the loss between the image and the global prediction. Furthermore, once no more predictions were found, the error image (input image minus global prediction) was used as

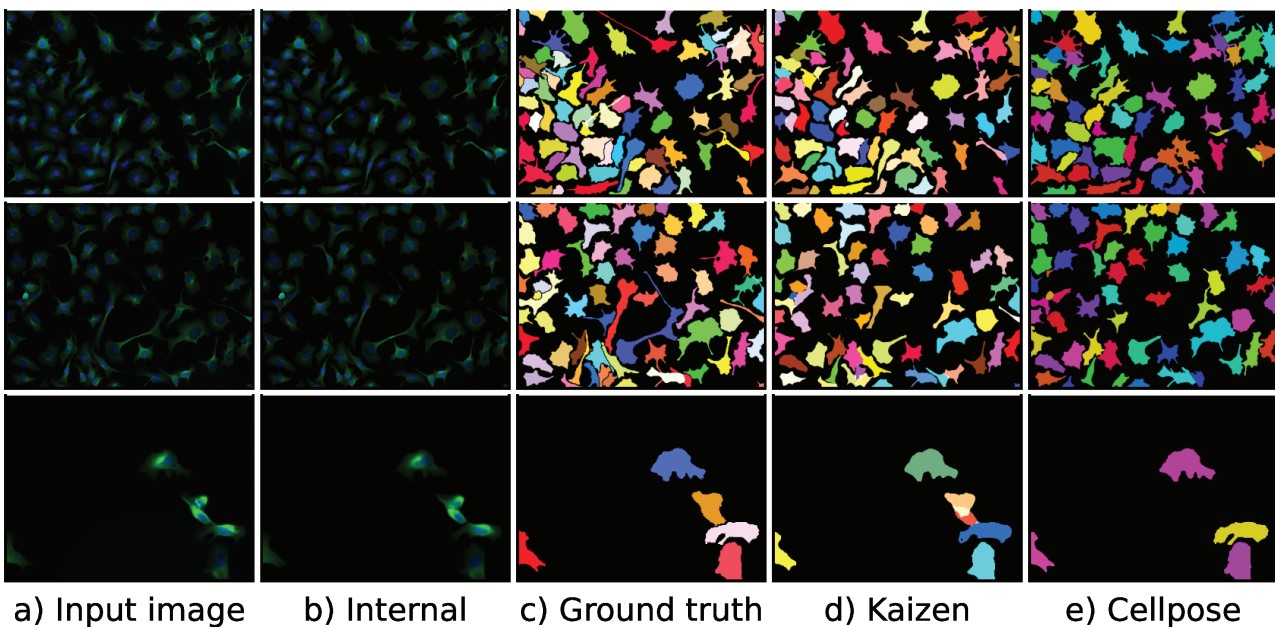

**a) Input image      b) Internal      c) Ground truth      d) Kaizen      e) Cellpose**

**Fig 5. Examples of *Kaizen* segmentation for neuroblastoma dataset.** The first two rows correspond to two dataset images, while the last row magnifies an image region. Panel a) depicts the original dataset image, and panel b) shows the corresponding internal image reconstruction created by merging the individual cells generated by the VQ-VAE. Panels d) and e) show Kaizen and Cellpose corresponding image segmentation. Panel c) depicts the ground truth segmentation.

new input to detect more cells and generate a second error image. This process was repeated several times to avoid missing cells.

Since typical models do not learn object representations, *Kaizen* was evaluated in two different instance segmentation datasets, showing competitive performance. More specifically, *Kaizen* obtained higher AP than Stardist across all thresholds, and higher $AP_{50}$, $AP_{60}$, and $AP_{70}$ than Cellpose.

In future work *Kaizen* can be extended by allowing several models to make simulataneous predictions. Adding evolutionary algorithms and more powerful generative models should improve further *Kaizen* performance.

## Author contributions

**Conceptualization:** Daniel Majoral.

**Formal analysis:** Marharyta Domnich.

**Investigation:** Daniel Majoral, Marharyta Domnich.

**Methodology:** Daniel Majoral, Marharyta Domnich.

**Software:** Daniel Majoral.

**Validation:** Marharyta Domnich.

**Visualization:** Daniel Majoral.

**Writing – original draft:** Daniel Majoral.

**Writing – review & editing:** Daniel Majoral, Marharyta Domnich.

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
