## [Decision Letter · Decision Letter 0]

28 Nov 2024

PONE-D-24-48691Kaizen: Decomposing cellular images with VQ-VAEPLOS ONE

Dear Dr. Majoral,

Thank you for submitting your manuscript to PLOS ONE. After careful consideration, we feel that it has merit but does not fully meet PLOS ONE’s publication criteria as it currently stands. Therefore, we invite you to submit a revised version of the manuscript that addresses the points raised during the review process.

We look forward to receiving your revised manuscript.

Kind regards,

Zeheng Wang

Academic Editor

PLOS ONE

Journal Requirements: When submitting your revision, we need you to address these additional requirements. 1. Please ensure that your manuscript meets PLOS ONE's style requirements, including those for file naming. The PLOS ONE style templates can be found at https://journals.plos.org/plosone/s/file?id=wjVg/PLOSOne_formatting_sample_main_body.pdf and https://journals.plos.org/plosone/s/file?id=ba62/PLOSOne_formatting_sample_title_authors_affiliations.pdf 2. Please note that PLOS ONE has specific guidelines on code sharing for submissions in which author-generated code underpins the findings in the manuscript. In these cases, we expect all author-generated code to be made available without restrictions upon publication of the work. Please review our guidelines at https://journals.plos.org/plosone/s/materials-and-software-sharing#loc-sharing-code and ensure that your code is shared in a way that follows best practice and facilitates reproducibility and reuse. 3. Thank you for stating the following in the Acknowledgments Section of your manuscript: "This research was supported by the Estonian Research Council Grants PRG1604, the European Union’s Horizon 2020 Research and Innovation Programme under Grant Agreement No. 952060 (Trust AI), by the Estonian Centre of Excellence in Artificial Intelligence (EXAI), by the Estonian Ministry of Education and Research." We note that you have provided funding information that is not currently declared in your Funding Statement. However, funding information should not appear in the Acknowledgments section or other areas of your manuscript. We will only publish funding information present in the Funding Statement section of the online submission form. Please remove any funding-related text from the manuscript and let us know how you would like to update your Funding Statement. Currently, your Funding Statement reads as follows: "Estonian Research Council Grant PRG1604https://etag.ee/en/ European Union’s Horizon 2020 Research and Innovation Programme under GrantAgreement No. 952060https://research-and-innovation.ec.europa.eu/funding/funding-opportunities/funding-programmes-and-open-calls/horizon-2020_en Estonian Centre of Excellence in ArtificialIntelligence (EXAI)https://exai.ee/ Estonian Ministry of Education and Researchhttps://www.hm.ee/en The funders did not play any role in the study design, data collection and analysis, decision to publish, or preparation of the manuscript" Please include your amended statements within your cover letter; we will change the online submission form on your behalf. 4. We note that your Data Availability Statement is currently as follows: All relevant data are within the manuscript and its Supporting Information files. Please confirm at this time whether or not your submission contains all raw data required to replicate the results of your study. Authors must share the “minimal data set” for their submission. PLOS defines the minimal data set to consist of the data required to replicate all study findings reported in the article, as well as related metadata and methods (https://journals.plos.org/plosone/s/data-availability#loc-minimal-data-set-definition). For example, authors should submit the following data: - The values behind the means, standard deviations and other measures reported;- The values used to build graphs;- The points extracted from images for analysis. Authors do not need to submit their entire data set if only a portion of the data was used in the reported study. If your submission does not contain these data, please either upload them as Supporting Information files or deposit them to a stable, public repository and provide us with the relevant URLs, DOIs, or accession numbers. For a list of recommended repositories, please see https://journals.plos.org/plosone/s/recommended-repositories. If there are ethical or legal restrictions on sharing a de-identified data set, please explain them in detail (e.g., data contain potentially sensitive information, data are owned by a third-party organization, etc.) and who has imposed them (e.g., an ethics committee). Please also provide contact information for a data access committee, ethics committee, or other institutional body to which data requests may be sent. If data are owned by a third party, please indicate how others may request data access.

**Additional Editor Comments:**

kindly follow the instructions of PLoS ONE publication to make the code and source data open-accessed.

Reviewers' comments:

Reviewer's Responses to Questions

**Comments to the Author**

1. Is the manuscript technically sound, and do the data support the conclusions?

Reviewer #1: Yes

Reviewer #2: Yes

2. Has the statistical analysis been performed appropriately and rigorously? 

Reviewer #1: Yes

Reviewer #2: No

3. Have the authors made all data underlying the findings in their manuscript fully available?

Reviewer #1: Yes

Reviewer #2: No

4. Is the manuscript presented in an intelligible fashion and written in standard English?

Reviewer #1: Yes

Reviewer #2: Yes

5. Review Comments to the Author

Reviewer #1: This paper introduces an innovative method for segmenting microscopy images into individual cells using a Vector Quantised-Variational AutoEncoder (VQ-VAE). Central to the work is the Kaizen approach, inspired by the brain's predictive coding, which enables object-level decisions for image decomposition. Experiments on two fluorescence microscopy datasets highlight improved separation of nuclei and neuronal cells in dense cell culture images.

In general, the proposed method has some novelty and achieved good performance. However, many deficiencies in this paper need to be improved:

1. The paper lacks a comprehensive comparison with state-of-the-art methods. While results are compared with Stardist and Cellpose, it is unclear how Kaizen performs against other recent advancements in cell segmentation, including promising deep learning models.

2. The potential overfitting of VQ-VAE to the specific datasets used is a concern. The paper does not adequately address whether Kaizen generalizes to other types of microscopy images or datasets, a critical factor for broader applicability.

3. The analysis focuses on average precision but lacks details on error types (e.g., false positives, false negatives) and their impact on downstream applications. A deeper error analysis could better highlight Kaizen's robustness.

4. While the method improves segmentation, the paper could strengthen its impact by discussing potential clinical applications or benefits for cell and tissue biology.

5. Although datasets are well-documented, the experimental setup and VQ-VAE training parameters are insufficiently detailed, hindering reproducibility by other researchers.

Reviewer #2: The paper introduces a promising approach to cellular image segmentation and achieves competitive results. However, to fully demonstrate the robustness and reproducibility of the method, the authors should address the detailed concerns about algorithm description, results presentation, and code availability.

1. Algorithm Description Lacks Sufficient Detail

1.1 Training Dataset Size and Splits

• The paper does not provide a clear description of the training dataset size and how the training, validation, and test sets are split. For reproducibility and proper evaluation of generalizability, it is important to include these details. Were cross-validation techniques employed? If not, what measures were taken to ensure the model is not overfitting?

1.2 Impact of Patch Size and Handling Cell Size Variations

• The choice of patch size (e.g., 40×40 for U2OS and 120×120 for Neuroblastoma) is mentioned but lacks justification. How was the patch size determined, and what is its impact on performance?

• Given the diverse sizes of cells within microscopy images, how does the model handle cells significantly smaller or larger than the chosen patch size? Does it involve any resizing, padding, or multi-scale training strategies?

1.3 Dataset Preprocessing Details

• Additional details on dataset preprocessing and any augmentation strategies employed during training would further facilitate reproducibility.

1.4 Error Image Generation and Point Selection

• How are the parameters for "distant points" determined, for example, why was a 7×7 kernel chosen for the convolution? Is it the same for different dataset.

• What is the rationale for selecting ten points for U2OS and one for Neuroblastoma datasets?

1.5 Impact of Multiple Iterations

• The paper mentions multiple iterations to refine predictions but does not discuss convergence criteria.

o In what scenarios does the iterative process converge quickly, and in what situations does it lead to diminishing returns or negative effects?

o A comparative analysis of convergence speed and computational efficiency with other segmentation methods would strengthen the evaluation.

2. Result Presentation and Comparison

• Inclusion of Comparative Visual Results, Figures 3 and 4 showcase Kaizen's results but lack comparative outputs from other methods. Including these would allow readers to visually assess the strengths and weaknesses of Kaizen relative to existing approaches.

3. Code and Reproducibility

• To support reproducibility and adoption, the authors should clarify whether they intend to release their code, pre-trained models, and a detailed pipeline for both datasets.

6. PLOS authors have the option to publish the peer review history of their article (what does this mean?). If published, this will include your full peer review and any attached files.

Reviewer #1: No

Reviewer #2: No

---

## [Author Response · Author response to Decision Letter 1]

19 Feb 2025

Response reviewer 1

We thank the reviewer for the time and effort in giving us constructive feedback. In the following we address each of the reviewer’s questions:

1. We compare Kaizen to two other methods, like in the literature, and we choose Stardist and Cellpose due to wide employment by researchers and code availability. Rather than reach SOTA performance we were interested in applying generative models to segmentation. However, we will make the code available to facilitate comparison with other methodologies.

2. We apply the methodology of the same model with token parameter changes to two different datasets, which shows some generalization capability. Rather than develop a SOTA method for a large amount of data we were more interested in exploring methods to apply generative models to segmentation. However, we agree with the reviewer and future work is needed with a more powerful generative model that can benefit greatly from larger data from different datasets.

3. We have added more detail regarding error types for the U20S dataset in the results section (line 185). For the neuroblastoma dataset, the results are more balanced with 41 false positives and 44 false negatives, and might not interest the reader.

4. Our background is mainly in computer science. We are hesitant to discuss clinical applications or benefits for cell and tissue biology, given our lack of experience and shallow knowledge on the topic.

5. The other reviewer also raised this issue about sharing the code and reproducibility by other researchers. We have decided to make the code available to facilitate reproducibility by other researchers. We are still working on refactoring the code since it was very messy. We will make everything available in github (https://github.com/Danielmaj/Kaizen) and in one of the repositories recommended by PLOS ONE.

Response reviewer 2

We sincerely thank the reviewer for raising critical questions and for providing comments to improve the manuscript.

1.1 We have added in section 2.4 a more detailed description of the dataset splits. No cross validation was performed in the dataset. Through experimentation we have found that when the vq-vae model overfits the dataset the final performance of the method is severely impaired. Thus the vq-vae is trained a small amount of epochs to avoid overfitting.

1.2 The patch size was chosen such that it covers the bigger cells. Patch size significantly affects performance. A smaller patch size will fail to segment big cells, in contrast a bigger patch size diminishes the vq-vae and segmentation accuracy. In the case of the neuroblastoma dataset there are a few cells four or five times larger than typical ones. The current implementation does not handle these large cells since increasing the patch size so much will diminish the precision for the average cell. We plan to address this issue in future work.

We approach this article as a proof of concept, we are not trying to achieve sota performance. We did not implement any approach to improve handling of different patch sizes.

1.3 The only data preprocessing performed was to normalize the images. No data augmentation was applied. We have updated section 2.6 with this information.

1.4 The idea behind the 7x7 kernel is two fold: Avoid predicting isolated noisy pixels and avoid predicting exactly at the cell’s boundary. The kernel parameter was selected by tests on the validation set of the U2OS dataset, but differences were minimal to slight changes in the parameter. The parameter is the same for both datasets.

The rationale for selecting ten points for U2OS and one for Neuroblastoma datasets is related to the convergence of the algorithm (see also 1.5 response), and memory-speed tradeoff. For the USO2 dataset keeping the parameter high helps the algorithm to not stop before predicting almost all cells in image and perform faster. However, increasing the parameter for the Neuroblastoma dataset might cause memory problems since the patches are bigger and parallel processing increases memory consumption. Increasing the parameter in this case did not improve convergence, thus we decided to keep it at one .

1.5 The algorithm time is proportional to the number of cells in the image. For empty images is almost instantaneous since it will only convolve a kernel of ones along the image and try to do a limited number of predictions. Thus the algorithm is very fast for very big images with low density of cells.

In general, very high density of cells, noisy images with a lot of artifacts, and difficult to predict cells for the model will increase the computational time, but in these cases it seems reasonable to spend more computation.

See attached figure with the time Kaizen takes to process an image respect to the number of nuclei in the image for the U2OS dataset

One advantage of the algorithm is a smaller memory requirement than other methods, since at any given time only part of the image is processed through the network.

We have considered adding a figure to the manuscript about this topic and discussing it. However, we thought that perhaps it is not a critical topic and we did not want to distract from the main message. Nevertheless, we are open to suggestions on tackling this topic in the manuscript.

2 We have added to Figures 3 and 4 outputs from other methods as suggested, for visual comparison.

3 The issue about reproducibility by other researchers was also raised by the other reviewer. We have decided to make the code available to facilitate reproducibility by other researchers. We are still working on refactoring the code since it was very messy. We will make everything available in github (https://github.com/Danielmaj/Kaizen) and in one of the repositories recommended by PLOS ONE.

---

## [Decision Letter · Decision Letter 1]

10 Mar 2025

PONE-D-24-48691R1Kaizen: Decomposing cellular images with VQ-VAEPLOS ONE

Dear Dr. Majoral,

Thank you for submitting your manuscript to PLOS ONE. After careful consideration, we feel that it has merit but does not fully meet PLOS ONE’s publication criteria as it currently stands. Therefore, we invite you to submit a revised version of the manuscript that addresses the points raised during the review process.

We look forward to receiving your revised manuscript.

Kind regards,

Zeheng Wang

Academic Editor

PLOS ONE

Journal Requirements:

Reviewers' comments:

Reviewer's Responses to Questions

**Comments to the Author**

1. If the authors have adequately addressed your comments raised in a previous round of review and you feel that this manuscript is now acceptable for publication, you may indicate that here to bypass the “Comments to the Author” section, enter your conflict of interest statement in the “Confidential to Editor” section, and submit your "Accept" recommendation.

Reviewer #1: (No Response)

Reviewer #2: All comments have been addressed

2. Is the manuscript technically sound, and do the data support the conclusions?

Reviewer #1: Yes

Reviewer #2: Yes

3. Has the statistical analysis been performed appropriately and rigorously? 

Reviewer #1: Yes

Reviewer #2: Yes

4. Have the authors made all data underlying the findings in their manuscript fully available?

Reviewer #1: Yes

Reviewer #2: Yes

5. Is the manuscript presented in an intelligible fashion and written in standard English?

Reviewer #1: Yes

Reviewer #2: Yes

6. Review Comments to the Author

Reviewer #1: The Kaizen method introduces a novel approach to learning object representations in microscopy images, drawing inspiration from human perception and predictive coding. Experiments show the effectiveness of the proposed method. To improve the paper, the following are suggested:

1. The method section lacks clarity. Adding symbols, formulas, and a framework diagram would better illustrate the training process.

2. Responses to both reviewers should be included in the paper, particularly regarding the algorithm's time aspect.

Reviewer #2: (No Response)

7. PLOS authors have the option to publish the peer review history of their article (what does this mean?). If published, this will include your full peer review and any attached files.

Reviewer #1: No

Reviewer #2: No

---

## [Author Response · Author response to Decision Letter 2]

10 Apr 2025

We sincerely thank the reviewer for providing comments to improve the manuscript.

1. We agree with the reviewer that the methodology section lacks clarity. We added a summary at the beginning of methodology, referencing figure 1 that tries to illustrate the method to facilitate reader understanding :

“An Illustration of Kaizen is shown in Figure 1. Kaizen uses a VQ-VAE model trained on microscopy images to predict one individual cell from an image with multiple cells. During inference the VQ-VAE iteratively predicts individual cells in the input microscopy image (Figure 1A). Kaizen maintains an internal predicted image formed by all the cells predicted so far (Figure 1B). The difference between the internal predicted image and the external image is the error image (Figure 1C). Kaizen accepts a new prediction only when it reduces the error, making the external image and the internal prediction more similar. Furthermore, the new predictions are made on regions with higher error (Figure 1C crosses), avoiding duplicate predictions. The process is repeated until the method is unable to predict new cells. Kaizen components are described in more detail below. “

We added a comment about the training process to increase clarity:

“The purpose of the training was for the VQ-VAE to produce a single cell image as output when given an image containing multiple cells.”

Regarding the framework diagram we were unable to design one to showcase the method. We will appreciate advice on how to draw one that conveys the method.

2. We have added a new figure illustrating the algorithm's runtime and a corresponding discussion in the results section.

“The impact on the algorithm of variations across entire images was also analyzed. As illustrated in Figure 4, with a fixed number of parallel predictions the processing time of Kaizen scales proportionally with the number of cells present in the image. For empty images, the computation is nearly instantaneous, as the algorithm primarily involves convolving a kernel of ones across the image and performing a limited number of predictions. Consequently, Kaizen remains highly efficient for large images with a low cell density. However, computational time increases in cases of high cell density, images with significant noise and artifacts, or instances where the model encounters cells that are challenging to predict”

Additionally, in the implementation section, we have provided a justification for using the kernel:

“This process aims to minimize the occurrence of predictions on background noise and cell boundaries”

---

## [Editor Report · Decision Letter 2]

11 Apr 2025

Kaizen: Decomposing cellular images with VQ-VAE

PONE-D-24-48691R2

Dear Dr. Majoral,

We’re pleased to inform you that your manuscript has been judged scientifically suitable for publication and will be formally accepted for publication once it meets all outstanding technical requirements.

Kind regards,

Zeheng Wang

Academic Editor

PLOS ONE
---

## [Editor Report · Acceptance letter]

PONE-D-24-48691R2

PLOS ONE

Dear Dr. Majoral,

I'm pleased to inform you that your manuscript has been deemed suitable for publication in PLOS ONE. Congratulations! Your manuscript is now being handed over to our production team.

Kind regards,

on behalf of

Dr. Zeheng Wang

Academic Editor

PLOS ONE